# High Levels of Incidental COVID-19 Infection in Emergency Urology Admissions: A Propensity Score-Matched Real World Data Analysis across Surgical Specialties

**DOI:** 10.3390/v16091402

**Published:** 2024-08-31

**Authors:** Alex Qinyang Liu, Eric Ka-Ho Choy, Peter Ka-Fung Chiu, Chi-Hang Yee, Chi-Fai Ng, Jeremy Yuen-Chun Teoh

**Affiliations:** 1S.H. Ho Urology Centre, Department of Surgery, Prince of Wales Hospital, The Chinese University of Hong Kong, Hong Kong, China; ericchoyckh@gmail.com (E.K.-H.C.); peterchiu@surgery.cuhk.edu.hk (P.K.-F.C.); yeechihang@surgery.cuhk.edu.hk (C.-H.Y.); ngcf@surgery.cuhk.edu.hk (C.-F.N.); 2Li Ka Shing Institute of Health Sciences, The Chinese University of Hong Kong, Hong Kong, China; 3Department of Urology, Medical University of Vienna, 1090 Vienna, Austria

**Keywords:** COVID-19, surgery, urology, acute urinary retention, urinary tract infection

## Abstract

**Background:** An incidental COVID-19 infection is often found in patients admitted for non-COVID-19-related conditions. This study aims to investigate the incidence of COVID-19 infections across surgical specialties including urology, general surgery, and orthopaedic surgery. **Methods:** This is a retrospective cohort study based on a territory-wide electronic database in Hong Kong. All emergency in-hospital admissions under the urology, general surgery, and orthopaedic surgery divisions in the public healthcare system in Hong Kong from January to September 2022 were included. All patients were routinely screened for SARS-CoV-2, based on admission protocols during the investigation period. Baseline characteristics were retrieved, with 1:1:1 propensity score matching being performed. Incidental COVID-19 rates were then compared across specialties. **Results:** A total of 126,034 patients were included. After propensity score matching, the baseline characteristics were well balanced, and 8535 patients in each group were analysed. Urology admission was noted to have a statistically significant higher incidence of incidental COVID-19 at 9.3%, compared to general surgery (5.4%) or orthopaedic surgery (5.6%). Amongst urology patients with incidental COVID-19 infection, 35.8% were admitted for retention of urine, 27.9% for haematuria, and 8.6% for a urinary tract infection. **Conclusions:** This large-scale cohort study demonstrated that incidental COVID-19 rates differ between surgical specialties, with urology having the highest proportion of incidental COVID-19 infection.

## 1. Introduction

The coronavirus disease 2019 (COVID-19) pandemic has significantly impacted healthcare systems worldwide, leading to unprecedented challenges and disruptions. It was declared a global emergency by the World Health Organization [1] and has affected an unprecedented 600 million people globally as of January 2023 [2]. Apart from acute admissions for symptomatic COVID-19, usually presenting with respiratory symptoms, the clinical presentation of COVID-19 can vary from asymptomatic infection to extra-respiratory symptoms [3].

The current literature suggests that up to 26% of patients were found to have incidental COVID-19 during in-hospital admissions [4]. These patients were often asymptomatic, especially in terms of the respiratory symptoms characteristic of COVID-19. This is of clinical concern as the undiagnosed COVID-19 status of these patients could translate into more lenient infection control measures. This, coupled with the highly contagious nature of the causative virus (severe acute respiratory syndrome coronavirus-2; SARS-CoV-2), could, in turn, expose unsuspecting healthcare staff and other inpatients to COVID-19 transmission during an in-hospital stay. Surgical specialties often involve close patient–staff interactions and necessitate the use of specialized equipment and environments, which can be conducive to the nosocomial transmission of COVID-19 [5,6].

In addition, patients presenting with extra-respiratory symptoms of COVID-19 may also be incorrectly admitted under non-medical specialties due to their symptoms masquerading as another virus. In urology, prostate epithelial cells have been suggested to be targets of SARS-CoV-2 [7] due to the co-expression of ACE2 and TMPRSS2. Apart from the ACE2 pathway, a systematic review has also revealed other pathways including androgen receptor-dependent TMPRSS2 expression, inflammation cascade, or metabolic dysregulation as a possible means through which the prostate is affected by SARS-CoV-2. Recent clinical studies have also demonstrated an increased incidence of urinary retention, haematuria, and UTIs in COVID-19 patients [8]. With predominant urological symptoms, patients with COVID-19 are more likely to be admitted to urology services, which may delay timely management of COVID-19. Similar trends in general surgical conditions have not been reported.

As a result of the above correlations between urological conditions and COVID-19, it is postulated that incidental COVID-19 incidence would be higher in urology emergency admissions compared to the other surgical specialties. One clinical implication is the higher risk of infection to healthcare workers and urologists involved in the management of emergency urology admissions, especially during a pandemic. Therefore, the objective of this study is to understand the incidence of COVID-19 infections across surgical specialties and explore if urology emergency admissions are associated with a higher incidence of incidental COVID-19. Firstly, it helps with the identification of any risk factors for incidental COVID-19 infection that would allow improved patient and staff protection. Secondly, it helps with assessing the effectiveness of current infection control protocols and identifying potential gaps. Thirdly, it contributes to the broader public health strategy by highlighting areas that require enhanced protective measures. The findings will also allow clinicians to draw correlation and association between COVID-19 and clinical symptoms to facilitate future investigation on the extra-respiratory manifestation of COVID-19. The results will shed light and provide reference for any future respiratory infectious disease outbreaks.

## 2. Methodology

### 2.1. Study Design

This is a retrospective cohort study comparing the incidence of incidental COVID-19 across selected surgical specialties, including urology, general surgery, and orthopaedic surgery. Data for this study were sourced from the territory-wide electronic patient record database, Clinical Data Analysis and Reporting System (CDARS), managed by the Hong Kong Hospital Authority. The Hospital Authority is the primary publicly funded healthcare provider for Hong Kong’s 7.4 million residents, covering up to 94% of all secondary and tertiary healthcare services in Hong Kong [9]. CDARS includes comprehensive clinical information such as patient demographics, survival data, hospitalization records, outpatient clinic attendance, diagnoses based on the International Classification of Diseases, Ninth Revision (ICD-9), procedural records, laboratory results, and medication prescription details. The coding of this data is performed by clinicians and administrative staff during their routine clinical duties and is not part of the study process, with a reliable coding accuracy of up to 99% [10]. Patients fulfilling the inclusion and exclusion criteria were identified in CDARS and assigned an anonymous identifier, with their relevant clinical information extracted and analysed.

The study was conducted in accordance with the Declaration of Helsinki. Ethical approval was obtained before the commencement of the study, with approval granted by the Joint Chinese University of Hong Kong–New Territories East Cluster Clinical Research Ethics Committee (CREC Reference Number 2022.319).

### 2.2. Patients

All patients admitted from the Accident and Emergency Department into selected surgical specialties including urology, general surgery, and orthopaedic surgery from January to September 2022 were included in this study. The scope of the study is territory-wide, including all 18 public hospitals in Hong Kong with emergency admission service. Patients < 18 years old were excluded from this study.

### 2.3. Outcomes

The primary outcome of the study is the identification of an incidental COVID-19 diagnosis defined by positive polymerase chain reaction (PCR) test results for SARS-CoV-2, as extracted from CDARS. During the height of the pandemic, within the investigation period, the emergency admission in the public healthcare system in Hong Kong adopted a very stringent SARS-CoV-2 screening and testing policy, with all inpatients receiving SARS-CoV-2 screening using a rapid antigen test (RAT). A positive RAT result was subsequently confirmed with a PCR test. The secondary outcome includes the descriptive analysis of principal diagnoses across different specialties.

### 2.4. Statistical Methods

Key co-morbidities of the included patients were extracted using the corresponding ICD-9 codes from CDARS and selected as covariates. Propensity scores of each patient with confounding covariates were calculated using a logistic regression model. Propensity score matching comparing urology against general surgery, as well as against orthopaedic surgery with a narrow caliper of 0.05 and 1:1 match ratio was performed to adjust and balance potential confounding covariates, including age, diabetes mellitus, hypertension, dyslipidemia, obesity, ischemic heart disease, stroke, smoking and related diseases, and alcohol-related diseases. The primary outcomes were compared using a chi-squared test. SPSS Statistics (version 25.0. Armonk, NY, USA: IBM Corp) and RStudio (build 576. PBC, Boston: RStudio team) were used for the data analyses in this study.

## 3. Results

A total of 126,034 patients from January to September 2022 were admitted via the emergency service to the 18 acute hospitals in Hong Kong. All eligible patients were included, amongst whom 8535 patients were admitted under urology, 70,061 patients were admitted under general surgery, and 47,438 patients were admitted under orthopaedic surgery. All patients were screened for SARS-CoV-2 after admission to a surgical ward. The baseline demographics and prevalence of co-morbidities are illustrated in Table 1. Urology patients were noted to be older in age, have a higher male-to-female ratio, and have a higher prevalence of all co-morbidities (except obesity and alcoholism) compared to general surgery and orthopaedic surgery.

Of the 126,034 patients, 6346 (5.04%) had incidental SARS-CoV-2 findings upon hospitalization. Urology had a statistically significant higher (*p* < 0.001) proportion of incidental SARS-CoV-2 findings (9.3%) than general surgery (4.8%) and orthopaedic surgery (4.7%). Subgroup analysis was performed based on sex and age, respectively. After stratification by sex, the incidental SARS-CoV-2 rate in urology patients was 9.7% and 8.4% for men and women, respectively, showing a consistently higher statistical significance than the sex-matched patients in general surgery (5.1% and 4.4% in male and female, respectively) and orthopaedic surgery (4.5% and 4.8% in male and female, respectively). Patients were also stratified by age groups, including ≤50 years old, 51 to 70 years old, and >70 years old. Among urology admissions, the rates of incidental SARS-CoV-2 were only 3.7% in patients ≤ 50 years old, 4.6% in patients between 51 to 70 years old, and 10.4% in patients > 70 years old. There were no statistically significant differences between urology and surgery and orthopaedic surgery in patients ≤ 50 years old (3.7% vs. 3.0% vs. 1.6%, *p* = 0.139); however, a significantly higher SARS-CoV-2 incidence was observed in urology patients aged 51 to 70 (4.6% vs. 2.8% vs. 2.3%, *p* < 0.001) and patients aged > 70 (10.4% vs. 5.4% vs. 5.6%, *p* < 0.001).

After the propensity score matching was adjusted to the patients’ demographics (age and gender) and co-morbidities (diabetes mellites, hypertension, hyperlipidaemia, obesity, ischemic heart disease, stroke, smoking, and alcoholic consumption), the baseline characteristics showed a good balance, as demonstrated in Table 2. The incidental SARS-CoV-2 rate in the urology patients remained significantly higher than general surgery and orthopaedic surgery patients, respectively, even after propensity score matching, as summarised in Table 3.

In those patients with incidental SARS-CoV-2, the most common diagnoses in urology upon admission were acute retention of urine (284, 35.8%), haematuria (221, 27.9%), and urinary tract infections (68, 8.6%). Abdominal pain (339, 10.2%), rectal bleeding (158, 4.8%), and epigastric pain (135, 4.1%) were the top three most common diagnoses in the general surgery admissions. In the orthopaedic emergency admissions, the top three diagnoses were closed femur transcervical fracture (185, 8.3%), back pain (182, 8.2%) and closed femur trochanteric fracture (177, 8.0%). The top ten diagnoses in each of the specialties are documented in Table 4.

## 4. Discussion

The data showed that, in our cohort consisting of 126,034 patients, 5.1% had an incidental SARS-CoV-2 infection after admission to surgical wards. Klann et al. found a rate of incidental SARS-CoV-2 infection of 26% in 1123 patients from March 2020 to August 2021 [4]. In a recent publication by McAlister et al., the rate of incidental SARS-CoV-2 infection in Canada was 4.48% out of 82,965 patients from March 2020 to July 2022 [11]. Nikolla et al. estimated a 29.4% incidental SARS-CoV-2 rate from 153,325 patients across the US from April 2020 to August 2023 [12]. Our data were collected from January to September 2022, at which time the major variant in Hong Kong was Omicron. The Omicron variant caused fewer low respiratory tract symptoms and less need for admission to hospital than for those infected with earlier variants [13]. Our result was largely compatible with McAlister et al., using a similar database and time frame. This is the largest study demonstrating incidental SARS-CoV-2 rate from emergency admissions to specific surgical wards.

Our findings demonstrated that urology admissions (9.3%) had a significantly higher incidental SARS-CoV-2 rate than that in general surgery (4.8–5.4%) and O&T (4.7–5.6%) before and after propensity score matching according to demographics and co-morbidities. It was not driven by the difference in age, gender, prevalence of common co-morbidities like diabetes, hypertension, hyperlipidemia, or the other covariates included for propensity score matching. It suggested that upon incidental SARS-CoV-2 diagnoses, there were extra-respiratory manifestation of SARS-CoV-2 infection, and more as urological symptoms. In our database, the top three most common urological presentations to emergency admission were acute retention of urine (*n* = 284, 35.8%), haematuria (*n* = 221, 27.9%), and urinary tract infection (*n* = 68, 8.6%). Our previous study demonstrated that SARS-CoV-2 is associated with an increased incidence of urinary retention, haematuria, and urinary tract infection regardless of severity [8], and the results of this study are in line with our previous report. With more urological manifestations of COVID-19, these infectious patients may be inappropriately admitted under urology care due to the apparent clinical presentation. Urological presentation aside, clinicians should also pay attention to gastrointestinal tract symptoms, such as abdominal pain, rectal bleeding, epigastric pain, as common presentation in our database. Balaphas et al. found that more than one-fifth of patients admitted for COVID-19 presented with abdominal pain. Epigastric pain is associated with dyspnea and a higher risk of adverse outcomes [6]. For orthopaedic presentation, an increase in hip fracture patients with concomitant COVID-19 infection was observed and associated with a significantly higher mortality rate than control in both Asian and Caucasian population [14,15,16,17].

In this study, we did not investigate the correlation between incidental infection and extra-respiratory presentation. A large population study to establish a correlation between incidental SARS-CoV-2 infection and extra-respiratory presentation, such as acute retention of urine or haematuria, is warranted. Nonetheless, clinicians should be aware of urological presentation as a possible extra-respiratory manifestation of SARS-CoV-2 infection, especially when no respiratory symptoms are reported, as it is associated with the highest proportion of incidental COVID infection in our database. Given these findings, the implementation of universal COVID-19 screening in urological patients should also be considered.

This study has several limitations. Firstly, the lack of randomization or blinding inevitably introduces bias to this retrospective study in nature. Propensity score matching has been used to eliminate statistical imbalance from baseline covariates, such as demographics and co-morbidities, to reach a fair comparison with minimized confounding factors. The scope of the CDARS database also limited the available covariates, with relevant conditions such as chronic kidney disease [18] not being included. Secondly, our data were collected from January to September 2022. A longer time frame and other SARS-CoV-2 variants were not included in this study. The nature of the routinely collected health data also means the data may suffer from issues like under-coding, especially for lifestyle risk factors such as smoking. Further study with a longer duration would be helpful to delineate the effect of different SARS-CoV-2 variants on incidental infection rate upon emergency admission to surgical wards specifically.

## 5. Conclusions

This is the largest cohort study concerning the incidental infection rate emergency in-hospital admissions across surgical specialties. Our study demonstrated that incidental COVID-19 rates differ between surgical specialties, with urology having the highest proportion of incidental COVID-19 infections compared to general surgery and orthopaedic surgery. There is a need for increased awareness in urological practice regarding the heightened risk of incidental COVID-19 infections and further considerations of the necessity of infection control and staff protection in urological emergencies.

## Figures and Tables

**Table 1 viruses-16-01402-t001:** Baseline characteristics.

	Urology (*n* = 8535)	General Surgery (*n* = 70,061)	*p*-Value	Orthopaedic Surgery (*n* = 47,438)	*p*-Value
Age (mean ± SD)	66.0 ± 18.4	63.2 ± 20.5	<0.001	62.9 ± 22.2	<0.001
Male (*n*, %)	6004 (70.3%)	37,520 (53.6%)	<0.001	22,965 (48.4%)	<0.001
Diabetes mellitus (*n*, %)	1733 (20.3%)	12,490 (17.8%)	<0.001	8534 (18.0%)	<0.001
Hypertension (*n*, %)	3367 (39.4%)	24,408 (34.8%)	<0.001	16,476 (34.7%)	<0.001
Hyperlipidemia (*n*, %)	1727 (20.2%)	12,578 (18.0%)	<0.001	8232 (17.4%)	<0.001
Obesity (*n*, %)	422 (4.9%)	3555 (5.1%)	0.605	2366 (5.0%)	0.866
Ischemic heart disease (*n*, %)	716 (8.4%)	5118 (7.3%)	<0.001	2962 (6.2%)	<0.001
Stroke (*n*, %)	728 (8.5%)	4736 (6.8%)	<0.001	2928 (6.2%)	<0.001
Smoking (*n*, %)	245 (2.9%)	1505 (2.1%)	<0.001	809 (1.7%)	<0.001
Alcoholism (*n*, %)	158 (1.9%)	1687 (2.4%)	0.001	953 (2.0%)	0.336

**Table 2 viruses-16-01402-t002:** Baseline characteristics after propensity score matching.

	Urology (*n* = 8535)	General Surgery (*n* = 8525)	Standardized Mean Difference	Orthopaedic Surgery (*n* = 8535)	Standardized Mean Difference
Age (mean)	66.0 ± 18.4	66.0	−0.001	66.0	−0.004
Male (*n*, %)	6004 (70.3%)	5989 (70.2%)	−0.004	5994 (70.2%)	0.003
Diabetes mellitus (*n*, %)	1733 (20.3%)	1673 (19.5%)	0.018	1739 (20.4%)	−0.002
Hypertension (*n*, %)	3367 (39.4%)	3377 (39.6%)	−0.002	3361 (39.4%)	0.001
Hyperlipidemia (*n*, %)	1727 (20.2%)	1743 (20.4%)	−0.005	1680 (19.7%)	0.014
Obesity (*n*, %)	422 (4.9%)	396 (4.64%)	0.014	345 (4.04%)	0.042
Ischemic heart disease (*n*, %)	716 (8.4%)	673 (7.89%)	0.018	671 (7.86%)	0.019
Stroke (*n*, %)	728 (8.5%)	686 (8.04%)	0.018	697 (8.17%)	0.013
Smoking (*n*, %)	245 (2.9%)	200 (2.44%)	0.023	221 (2.59%)	0.017
Alcoholism (*n*, %)	158 (1.9%)	142 (1.73%)	0.018	129 (1.51%)	0.025

**Table 3 viruses-16-01402-t003:** Incidental COVID-19 diagnosis across surgical specialties before and after propensity score matching.

	Urology (*n* = 8535)	General Surgery (*n* = 70,061)	Orthopaedic Surgery (*n* = 47,438)
Incidental COVID-19	793 (9.3%)	3329 (4.8%)	2224 (4.7%)
*p*-value *	N/A	<0.001	<0.001
After propensity score matching			
	**Urology (*n* = 8535)**	**General surgery (*n* = 8535)**	**Orthopaedic surgery (*n* = 8535)**
Incidental COVID-19	793 (9.3%)	461 (5.4%)	474 (5.6%)
*p*-value *	N/A	<0.001	<0.001

Continuous variables are presented as mean ± standard deviation. * Chi-squared test compared to urology.

**Table 4 viruses-16-01402-t004:** Top ten most common diagnoses in patients with incidental COVID-19 across surgical specialties.

	Urology (*n* = 793)	General Surgery (*n* = 3329)	Orthopaedic Surgery (*n* = 2224)
1	Acute retention of urine (*n* = 284, 35.8%)	Abdominal pain (*n* = 339, 10.2%)	Closed femur transcervical fracture (*n* = 185, 8.3%)
2	Haematuria (*n* = 221, 27.9%)	Rectal bleeding (*n* = 158, 4.8%)	Back pain (*n* = 182, 8.2%)
3	Urinary tract infection (*n* = 68, 8.6%)	Epigastric pain (*n* = 135, 4.1%)	Closed femur trochanteric fracture (*n* = 177, 8.0%)
4	Nephrostomy complications (*n* = 25, 3.2%)	Gastrointestinal haemorrhage (*n* = 114, 3.4%)	Hip joint pain (*n* = 69, 3.1%)
5	Loin pain-haematuria syndrome (*n* = 23, 2.9%)	Intestinal obstruction (*n* = 56, 1.7%)	Cellulitis and abscess of leg (*n* = 67, 3.0%)
6	Ureteric stone (*n* = 20, 2.5%)	Gastritis (*n* = 46, 1.4%)	Gangrene (*n* = 60, 2.7%)
7	Pyelonephritis (*n* = 19, 2.4%)	Acute cholecystitis (*n* = 39, 1.2%)	Closed distal radial fracture (*n* = 43, 1.9%)
8	Foley catheter complication (*n* = 16, 2.0%)	Choledocholithiasis (*n* = 29, 0.9%)	Knee pain (*n* = 39, 1.8%)
9	Scrotal/testicular pain (*n* = 12, 1.5%)	Constipation (*n* = 28, 0.8%)	Laceration (*n* = 28, 1.3%)
10	Prostatic hyperplasia (*n* = 9, 1.1%)	Duodenal ulcer (*n* = 27, 0.8%)	Decubitus ulcer (*n* = 26, 1.2%)

## Data Availability

De-identified data on individual participants and the study protocol will be available after publication up to 36 months, with investigators whose proposed use of the data has been approved by an independent review committee identified for this purpose to achieve aims in an approved and methodologically sound proposal. Proposals should be directed to Alex Qinyang Liu, with data requestors needing to sign a data access agreement.

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
