# Peer review of "High Levels of Incidental COVID-19 Infection in Emergency Urology Admissions: A Propensity Score-Matched Real World Data Analysis across Surgical Specialties"

_viruses, 2024, doi:10.3390/v16091402_

Round 1

Reviewer 1 Report

Comments and Suggestions for Authors

It is a interesting study and we are very pleased to see studies that assess the incidence of extra-expiratory COVID-19 manifestations, however:

1. We would like our authors to talk about the connection between epithelial cells from urinary tract and SARS-COV 2 spike protein receptor (briefly).

2. Among comorbidities we are surprised that chronic kidney disease was not included since there is a higher prevalence of CKD in hospitalized patients with COVID 19, how is that possible? for more information: https://doi.org/10.3892/etm.2022.11634 

3. Are any of the covariates associated with a positive PCR for SARS-COV2?

4. After matching the number positive PCR tests for SARS-COV 2 in urology did not change, could you explain why? detail a bit more about the matching decision? it is a bit unclear, and the tables should elaborate more on this!

5. Did the authors consider to take some of the urology-related symptoms in account as a covariate?

This study overall, is interesting, but it need to elaborate more on presentation of results and describing statistical analysis 

Comments on the Quality of English Language

English quality of language is appropriate, with minor adjustments

Author Response

Please see the attachment, thank you!

Reviewer 2 Report

Comments and Suggestions for Authors

The study topic is interesting. The biggest advantage of this study is the number of participants and methodology based on the big registry.

However, this study requires revisions:

1. Title

revision is needed - SARS-CoV-2 infection or COVID-19 (as a disease) 

Taking into account methodology, the study title should be revised to provide a more comprehensive overview. 

2. The abstract seems to be too long and does not match the criteria of the Viruses journal - please keep to the limit of the words. Please also clarify the text.

3. The Introduction section is well-prepared. The readers may benefit from 2-3 sentences on the study rationale - why urology? 

4. There is a lack of precisely defined study aims. Please provide a clear aim of the study.

5. Methods - this section requires major revisions

"126,034 patients were included for initial analysis, with 8535 in the urology group, 70061 in the general surgery group, and 47438 in the orthopedic surgery group" - if the authors have 126k records, it is unclear why they focused on urology rather than overall surgical admissions. 

it should be clearly defined how many hospital were indluded in this study. A brief characteristics of this hospital (and its relevance in the health system in Hong Kong) will be helpful.

It is unclear whether 126,034 patients were admitted in an emergency mode.

6. Results

this section is very short. This is unacceptable in the case of original articles. Please provide a more precise analysis (in subgroups by demographic, health status, etc.) or resubmit this study as a brief report.

7. Discussion

Please revise the first paragraph to summarize key findings (not to repeat to study aim).

please revise the limitations as this study has a lot of limitations

8. Conclusions are not fully supported by the results

In general, there is a lack of logical structure in the paper. The study aim is unclear. The scope of analysis is very limited and below the scientific level of Viruses Journal.

Comments on the Quality of English Language

Please double-check the text as some minor language erros are present in the text.

Author Response

Please see the attachment, thank you!

Round 2

Reviewer 1 Report

Comments and Suggestions for Authors

The article High incidental COVID-19 infection in emergency urology admissions: a propensity-score matched real world data analysis across surgical specialties tries to provide evidence for higher incidence of COVID-19 cases in urology department.

Recommendation:

1. I still see renal function, or at least the eGFR value relevant, therefore observation 2 was not fully assessed: https://doi.org/10.3892/etm.2022.11634

2. Since urological symptoms like hematuria could be present in COVID-19 clinical manifestation, why the authors not taken into account the fact that a higher incidence of these symptoms among all surgical patients would explain the higher incidence of COVID-19 patients in urology ward?

3. Most of the tables miss p-values for independence test or Kruskal-Wallis.

4. Hematuria and acute retention of urine is more prevalent in urologic patients, could the incidence of these symptoms be evaluated in COVID-19 cases across all surgical wards?

5. The supplementary table is irrelevant. 

6. Prevalence of some conditions is questionable (smoking only 2%) given the fact that the mean incidence is 10%; 

7. In table 4 the number of positive PCR test for SARS-COV 2 do not correspond to those in table 3. 

8. Not a single cancerous patient in these wards?

Author Response

Please see the attachment, thank you!

Reviewer 2 Report

Comments and Suggestions for Authors

The Authors addressed all the comments. 

Author Response

Thank you for your previous comments! Your inputs have helped us refine the article into a more accurate and readable form.